# Leaves and Fruits Preparations of *Pistacia lentiscus* L.: A Review on the Ethnopharmacological Uses and Implications in Inflammation and Infection

**DOI:** 10.3390/antibiotics10040425

**Published:** 2021-04-12

**Authors:** Egle Milia, Simonetta Maria Bullitta, Giorgio Mastandrea, Barbora Szotáková, Aurélie Schoubben, Lenka Langhansová, Marina Quartu, Antonella Bortone, Sigrun Eick

**Affiliations:** 1Department of Medicine, Surgery and Experimental Sciences, University of Sassari, Viale San Pietro 43, 07100 Sassari, Italy; 2C.N.R., Institute for Animal Production System in Mediterranean Environment (ISPAAM), Traversa La Crucca 3, Località Baldinca, 07100 Sassari, Italy; simonettamaria.bullitta@cnr.it; 3Department of Biomedical Sciences, University of Sassari, Viale San Pietro 43/C, 07100 Sassari, Italy; 30050744@studenti.uniss.it; 4Faculty of Pharmacy, Charles University, Akademika Heyrovského 1203, 50005 Hradec Králové, Czech Republic; szotakova@faf.cuni.cz; 5Department of Pharmaceutical Sciences, University of Perugia, Via Fabretti, 48-06123 Perugia, Italy; aurelie.schoubben@unipg.it; 6Institute of Experimental Botany, Czech Academy of Sciences, Rozvojová 263, 16502 Prague, Czech Republic; langhansova@ueb.cas.cz; 7Department of Biomedical Sciences, University of Cagliari, Cittadella Universitaria di Monserrato, 09042 Cagliari, Italy; quartu@unica.it; 8Dental Unite, Azienda Ospedaliero-Universitaria di Sassari, 07100 Sassari, Italy; antonella.bortone@aousassari.it; 9Department of Periodontology, School of Dental Medicine, University of Bern, Freiburgstrasse 3, 3010 Bern, Switzerland

**Keywords:** essential oils, terpenoids, polyphenols, water extracts, ethanol extracts, natural antimicrobials, natural anti-inflammatory, Mediterranean plants, pharmaceutical plants

## Abstract

There is an increasing interest in revisiting plants for drug discovery, proving scientifically their role as remedies. The aim of this review was to give an overview of the ethnopharmacological uses of *Pistacia lentiscus* L. (PlL) leaves and fruits, expanding the search for the scientific discovery of their chemistry, anti-inflammatory, antioxidative and antimicrobial activities. PlL is a wild-growing shrub rich in terpenoids and polyphenols, the oil and extracts of which have been widely used against inflammation and infections, and as wound healing agents. The more recurrent components in PlL essential oil (EO) are represented by α-pinene, terpinene, caryophyllene, limonene and myrcene, with high variability in concentration depending on the Mediterranean country. The anti-inflammatory activity of the oil mainly occurs due to the inhibition of pro-inflammatory cytokines and the arachidonic acid cascade. Interestingly, the capacity against COX-2 and LOX indicates PlL EO as a dual inhibitory compound. The high content of polyphenols enriching the extracts provide explanations for the known biological properties of the plant. The protective effect against reactive oxygen species is of wide interest. In particular, their anthocyanins content greatly clarifies their antioxidative capacity. Further, the antimicrobial activity of PlL oil and extracts includes the inhibition of *Staphylococcus aureus*, *Escherichia coli*, periodontal bacteria and *Candida* spp. In conclusion, the relevant scientific properties indicate PlL as a nutraceutical and also as a therapeutic agent against a wide range of diseases based on inflammation and infections.

## 1. Introduction

The undesirable side effects of antibiotics in addition to increasing microbial resistance have created a demand for new alternative molecules [1]. Non-steroidal anti-inflammatory drugs (NSAIDs) or even steroids, often inducing toxic side-effects [2] have also gained further interest as molecules presenting an anti-inflammatory character.

On these bases, there is an increasing attention towards revisiting plants for drug discovery, proving scientifically their role as popular remedies to diseases [3,4]. Approved therapeutic agents such as statins, tubulin-binding anticancer drugs and some types of immunosuppressants are examples of molecules originating from natural plants [5].

In the Mediterranean region, folk medicine has used extracts and oils derived, e.g., from *Malva* species [1], *Thymbra capitate* (L.) Cav. [2] and *Olea europea* L. [3] for many years.

The biological activity of essential oils (EOs) and polyphenols from plants and herbs is related to the presence of different chemical classes. In this regard, terpenes and terpenoids in EOs are promising agents in the prevention and treatment of diseases [6,7]. Terpenes are hydrocarbons, and terpenes containing additional functional groups, usually oxygen-containing, are called terpenoids [8]. They are lipophilic, and interact with cell membranes, neuronal and muscle ion channels, neurotransmitter receptors, G-protein coupled receptors, second messenger systems and enzymes [9]. Their beneficial effects and their roles have been evaluated for many decades in human disease, such as inflammatory diseases, tumorigenesis and neurodegeneration using cell and animal models, suggesting terpenes and terpenoids as potential chemopreventive and therapeutic agents [10]. Further interesting capabilities have been ascribed to polyphenols from botanical species, with molecules including tannins, flavonoids and lignin-carbohydrate complexes having been associated with strong antimicrobial, anti-inflammatory and antioxidant properties [11].

In this context, *Pistacia lentiscus* L. (PlL) is a wild-growing shrub rich in terpenoids and polyphenols [12]. PlL includes numerous wild and cultivated species, distributed in the Mediterranean and Middle Eastern areas. Although there has been broad investigation on the aromatic natural resin and its clinical application [13,14], the scientific data regarding PlL oil or extracts of leaves, fruits and woods have been not summarized yet. Today, the scientific interest in these edible and not-edible parts of PlL is wide-spreading, as some studies underlined the potential benefit against inflammation and infections [12,15,16]. Additionally, the high content of polyphenols found in the extracts make them attractive against chronic and degenerative diseases and as nutraceuticals in human health [15].

Given the above considerations, the purpose of this review was to screen the biological properties of PlL EO and extracts of leaves and fruits. Starting from this, we searched for the phytochemistry of PlL growing in the different geographical areas and reported on the anti-inflammatory and antimicrobial abilities of the plant. For that purpose, a search for the existing literature was made by using data bases such as MEDLINE/PubMed and the Cochrane Library electronic databases.

## 2. Botany and Taxonomy

### PlL Belongs to the Genus Pistacia, Anacardiaceae Family, Order Sapindales

Different classifications have been proposed regarding the *Pistacia* genus. One of the most known is that of Zohary [16], who classified the genus into four main groups according to the characteristics of the leaf and nut morphology (Table 1).

In the Mediterranean area (Middle East and Europe), the three most represented species are the following:*Pistacia vera* L., which is characteristic of the temperate areas of Asia Minor, and grown abundantly in Greece, the Aegean Islands and in Sicily (Italy). This species has been known since ancient times as attested in reports of the Old Testament. Additionally, there have been notices of *Pistacia vera* by Persian and Greek populations since the 6th and the 3rd century B.C., respectively.*Pistacia terebinthus* L., originating from the island of Chios, has spread to all the Mediterranean coasts over the centuries. Today, it is mostly present in Portugal, Palestine and North Africa, and in the Middle East of Asia till the western borders of India. In Italy, it is mainly found in the southern part of the peninsula and in Sardinia and Sicily.*Pistacia lentiscus* L., also known as mastic tree or lentisk (Figure 1).

PlL represents one of the most typical shrubs in the Mediterranean maquis (shrubland) of Europe, Morocco, Turkey, Iraq and Iran [17]. In Italy, it is characteristic of the sensitive ecosystem, like that of Sardinia [18], where it grows along the coast up to 700 m above sea level.

PlL is an evergreen environmentally sustainable shrub. It is well-adapted to harsh growing conditions, dryness and a warm environment, which all exercise an influence on the genotype and richness of secondary metabolites [19]. The plant is dioecious, where male and female flowers are on independent trees. The leaves are leathery, bright green and alternate. They are arranged in compound, pinnate whorls. The unisexual flowers are grouped in clusters. The globular fruit is a fleshy drupe, which ripens in August and ranges in color from red to brown in view of the different degrees of maturity [17]. PlL can develop leaf galls due to insect attack, particularly aphid attacks [20]. Common aphid species, such as *Slavum wertheimae* and *Baizongia pistaciae L*., manipulate the leaves to form tumorous galls for the safety and nutriment of their larvae [21]. The galls are rich in volatile-like terpenes with an abundance of monoterpenes, α -pinene and limonene [22]. Their chemical composition differs from that of the healthy leaves, which have in general a higher content of sesquiterpenes [22].

## 3. Historical and Cultural Use

PlL has had a wide range of applications over the centuries. One of the oldest dates back to the Nuragic civilization (1800 to 238 BCE) and was ascribed to the Sardinian population: the oil obtained by cold-pressing the berries was widely used for social purposes, i.e., home or votive lighting lamps, cooking, as well as as a popular remedy [17]. This habit is attested to by the presence of residues of “olium lentiscinum” often found during archaeological excavations in “torcularia” (ancient oil mills) [18].

Today, PlL is considered as an environment phytostabilizer due to the ability to detoxify the soil from harmful pollutants and heavy metals [23]. Furthermore, the plant represents an important source to increase milk quality and dairy products from ruminants browsing the Mediterranean maquis [24].

## 4. Ethnopharmacology

The ethnopharmacological survey on the medicinal use of PlL is reported in Table 2.

As previously referred to, there are few written indications on the medicinal use of the oil as a crude compound, or of the extracts as well. Drinking water extracts and topical application of the extracts or even whole parts of the plant (woods or leaves) have been the most common means of antagonizing gastrointestinal, hepatic, urinary, pulmonary and neurological diseases. In fact, the medicinal value of PlL in popular medicine covers a wide range of diseases, mainly including inflammatory processes and infections.

The Sardinian population has always found the medicinal properties of PlL highly appealing. A large number of publications report PlL oil and water extracts as useful means against a wide variety of inflammatory diseases, infections, allergies [21,27,28,32,33] and gastrointestinal disorders [28,29,30], and as wound healing [31]. It is further interesting to note that the Sardinian population administrated PlL as a smoke obtained by burning or boiling the soft wood and leaves, particularly in the cases of osteoarthritis, bronchitis and allergies [30]. In addition, PlL is still used as a remedy toward tooth ache and gingival inflammation by administering extracts from the leaves as oral mouthwash, beverages or by directly chewing the soft stems and leaves [30].

Similar beneficial effects have been reported by using the plants growing in the Southern (Campania) [32] and in the center (Abruzzo, Marche and Toscana) of Italy, in Tunisia [27] and in Spain [34].

PlL has been one of most used plants in Israeli and neighboring countries’ traditional pharmacology [40]. In Jordan, it was commonly used to antagonize jaundice [38,39]. In Algeria, it is known as antimicrobial, antioxidant, hypotensive and hypoglycemic agent [44]. In Morocco and Tunisia [35,37], PlL has been largely used as a remedy against gastrointestinal, kidney and hepatic disorders, in addition to being used to treat hypertension, diabetes, cardiac diseases, coughs, sore throats and eczema. Similar applications are reported in Turkey [43] and in Iran [41,42]. Meanwhile, in Tunisia, Spain and in the center of Italy, it has emerged as an agent against hypertension and heart diseases [27,37,41,42,43].

Regarding the veterinary use, in Sardinia, domestic animals are still treated by PlL wood to combat gastrointestinal disorders, and by swab bark in wound healing procedures and skin diseases [45]. Meanwhile, in Spain, the leaves are mentioned to treat specifically canine distemper [33].

## 5. Phytochemical Constituents

Mainly leaves and fruits are used for preparations of EO and also of water and alcoholic extracts. In Table 3, the major compounds of the EO and the extracts are listed. Hydro-distillation using Clevenger-type devices, and ethanol solvent have been the more common methods to obtain, respectively, the oil and extracts from leaves and fruits. However, the oil obtained by hydro-distillation and the extracts by solvents can have different organoleptic profiles and chemical compositions. These differences, in turn, will affect some properties, among which is the antimicrobial capacity, which is reported to be higher in a material solvent extract in comparison to a hydro-distilled [46]. Additionally, gas chromatography-mass spectroscopy (GC/MS) and high-performance liquid chromatography (HPLC) have been the most useful means to quantify phytochemically the oils and extracts, respectively [47].

PlL EO is constituted by a mixture of terpenes and terpenoids, mainly monoterpenes and sesquiterpenes, which are also responsible for the characteristic smell and flavoring of the plant [12,16,21,31]. It has been reported that terpenes in PlL are more genetically than environmentally related [48]. Nevertheless, the environment of growing, seasonability of harvesting and kind of material (edible or not-edible parts of the plant) have to be considered when explaining the differences in chemistry of the oils and extracts [19]. Up to 64 chemical constituents have been reported in the PlL EO fingerprint, in addition to other fractions that cannot be quantified by the assays [49]. Some of these terpenoids are constituent fractions of cannabis sativa [9], and called “non-cannabinoids terpenoids”. In PlL oil, non-cannabinoid terpenoids are more likely to be represented by α-pinene, myrcene, limonene, (E)-β-caryophyllene and γ- terpinene (Table 3). They are also included in the list of “terpene super classes” [9]. Furthermore, it is appealing to report that when non-cannabinoid terpenoids reach a concentration equal to or higher than 0.05% in an oil, they can confer pharmacological properties to such an oil, which can be classified as pharmaceutically active [50].

In view of the prevalent fractions of monoterpenes and oxygenated sesquiterpenes, the EO can be grouped into different chemotypes [17]. In this regard, the recurrent higher amount of α-pinene (16.9–19.5%) and terpinen-4-ol (7.7–16.5%) in comparison to the other compounds, allowed the classification of the oil from the leaves of PlL growing in Sardinia as the α-pinene/terpinen-4-ol chemotype [19,49]. Similarly, the Greek oil from PlL leaves is the α-pinene/terpinen-4-ol chemotype [51,51]. Nevertheless, the simultaneous existence of different chemotypes in a place can be justified by dissimilar geographical sites of harvesting in that country. An example is represented by the Corsican chemotype, which is expressed by three main phenotypes: the first is α-pinene/terpinen-4-ol; the second is terpinen-4-ol/limonene; and the third is myrcene-rich (88%) [19]. Very characteristic is the high content of δ-3-carene (65%) in the Egyptian oil [52], while the monoterpene terpinen-4-ol, together with α-pinene and the sesquiterpene myrcene, are among the higher represented fractions in the chemistry of PlL EO from Spain, Morocco and Turkey [53,54,55]. Conversely, α-pinene (65–86%) and β-myrcene (3%) are the major fractions which characterize the oils of mastic from plants growing in Spain [53].

Regarding the sesquiterpenes, limonene, α- and β-caryophyllene, D-germacrene, δ-cadinene and α-cadinol, β-bisabolene, β-bourbonene and caryophyllene oxide, they have shown extremely variable concentrations in PlL EO [16,21,53,56]. With respect to PlL leaves oil, the fruits oil changes significantly in its chemistry: limonene, sabinene and myrcene have been identified as the main representative fractions, in addition to α-pinene [56]. Additionally, the oil from the berries is rich in anthocyanins [57], which in addition to the concentration of fatty acids, such as oleic acid and linoleic acid [58], are precious as antioxidant compounds [59].

Furthermore, comprehensive studies have been conducted analyzing methanol and alcohol extracts of PlL leaves, while at the same time investigating their chemical profile, where a high concentration of phenolproponoids is reported (Table 3).

In this regard, an interesting study was carried out by Romani and co-workers [60]. Using ethyl acetate and methanol fractions of PlL leaves, the authors identified a high polyphenol content in the extracts, which represented 7.5% of the leaf dry-weight. In this content, three major classes of secondary metabolites were identified: (i) gallic acid and galloyl derivatives of both glucose and quinic acid; (ii) flavonol glycosides, i.e., myricetin and quercetin glycosides; and (iii) anthocyanins, namely delphinidin 3-O-glucoside and cyanidin 3-O-glucoside. All of these represent strong antioxidant polyphenols, which have implications in the prevention of chronic and inflammatory diseases [59]. Additionally, 46 different compounds were identified in the methanol extracts of PlL leaves from plants growing in Algeria [61].

Among the compounds in PlL extracts there were discovered to be relevant antioxidant agents, which may attest not only to the activity of PlL in preventing diabetic complications, cholesterol absorption and lipid metabolism [81,82], but also the remarkable capacity of PlL in managing intestinal inflammatory diseases as reported in the ethnopharmacological survey (Table 2). In fact, recent studies validated the capacity of polyphenols in managing the microbial and metabolomic patterns in the body [83,84,85,86]. Particularly, at the intestinal tract, these compounds can stimulate the multiplication of beneficial microorganisms and prevent the adhesion or directly disrupt the membrane ions flux of pathogens [87,88].

## 6. Anti-Inflammatory and Antioxidative Activities

As mentioned above, the anti-inflammatory effect of PlL is of high relevance in ethnopharmacology. The presence of important anti-inflammatory terpenes in the composition of PlL EO can explain its efficacy. It is well-demonstrated that terpenes are capable of inhibiting several inflammatory molecules, e.g., IL-1β, IL-6, TNFα and COX-2 [6,7,10], thus disrupting the amplification of inflammatory mechanisms. Meanwhile, the anti-inflammatory properties of PlL extracts can be related to the richness of polyphenols, the interplay of which in the inflammatory cascade is mainly demonstrated toward macrophages by inhibiting multiple key regulators of the inflammatory response [89]. Additionally, polyphenols reduce the release of arachidonic acid, prostaglandins and leukotrienes directly related to the inhibition of COX and LOX [90]. Other considerations regard several flavonoids in polyphenols, which can directly modulate the expression of pro-inflammatory cytokines and chemokines [91]. The ability of these natural compounds to modify the expression of several pro-inflammatory genes in addition to their antioxidant characteristics, such as reactive oxygen species (ROS) scavenging, contributes to the regulation of inflammatory signaling [92,93].

To clarify scientifically the anti-inflammatory character of PlL EO and extracts, they have undergone investigations by in vitro and in animal model studies with the intent to elucidate any selective interaction toward proteins and enzymes participating in the inflammatory pathway (Table 4).

### 6.1. Inhibitory Activity against Proinflammatory Cytokines and against Arachidonic Acid Cascade

Remila and co-workers [96] examined the anti-inflammatory activity of leaves and fruits extracts by measuring the secretion of IL-1β by macrophages exposed to adenosin triphosphate (ATP) or H_2_O_2_. The authors found PlL leaves extract significantly reduced the production of IL-1β from ATP- or H_2_O_2_-activated cells. The inhibitory capacity of the leaves extract was higher in comparison to that of the fruits and of quercetin and gallic acid (tested as isolated fractions of the polyphenol mixture). The data was explained by the higher content of the total phenols and flavonoids in the leaves compared to the fruits and by the synergy between the pharmacological biomolecules of PlL extract. Similar considerations and a dose-dependent anti-inflammatory effect of the leaves extract was reported in other studies, which further strengthened the capacity of flavonoids and tannins [97].

Comparable anti-inflammatory values were reported in regard to PlL EO using the carrageenan-induced paw edema and cotton pellet-induced granuloma in a rat model [29]. Particularly, it was evidenced that when applied topically, PlL EO from leaves significantly inhibited the development of granuloma and the serum level of TNF-α and IL-6 in reply to the irritants. The result was mainly related to the activity of α-pinene, β-pinene, α- phellandrene and sabinene, which were highly represented in the chemistry of the hydrodistilled oil.

Only one study has investigated the inhibitory activity of the whole PlL EO. It was obtained from the leaves of plants growing in Sardinia and tested against COXs and LOX [49]. The IC_50_ values were 10.3 ± 4.4 μg/mL and 6.1 ± 2.5 μg/mL PlL EO for COX-1 and COX-2, respectively, with higher inhibitory activity toward COX-2 in comparison to that produced towards COX-1. Additionally, COX-2 inhibition by the EO was similar to that recorded using ibuprofen as a positive control. The activity of the oil against LOX did not reach the IC_50_ value, as PlL EO lowered LOX activity by 30% compared to the control. Despite the low LOX inhibition, the study strengthens the oil as a potential dual inhibitory compound, which was intensively researched in pharmacology to antagonize a great number of inflammatory processes where these enzymes sustain and amplify the disease [68,78]. The data obtained in that investigation were addressed to the mixture of terpenoids comprising the oil, with particular regard to α-pinene and terpinen-4-ol (33.38%), enriched by the non-cannabinoid terpenoid limonene (3.4%), β-myrcene (0.9%), (Z)-caryophyllene (1.4%) and (E)-β-caryophyllene (0.1%). The mixture allowed the classification of the oil as pharmacologically active [50].

### 6.2. Inhibiting Activity against ROS Molecules

The protective effect of PlL EO and extracts against ROS has been deeply documented in the literature (Table 5). From a scientific point of view, this capacity can be related to the terpenes and polyphenols of the EO and of the extracts, respectively [10,59]. As the accumulation of ROS directly affects the healthy tissue systems, including cellular lipids, nucleic acids and proteins [98], the capacity of the EO and the extracts was studied directly using cell lines and intracellular ROS evaluation assays, or by chemical methods, particularly using the 2,20-azinobis(3-ethylbenzothiazoline-6-sulphonic acid) diammonium salt (ABTS) and the 1,1-diphenyl-2- picrylhydrazyl (DPPH) chemical assays, and more recently using electron chemical devices.

As it is shown in Table 4, the anti-ROS ability has been mainly investigated in PlL extracts. Scientifically, this ability has been addressed to the richness in polyphenols the extracts possess, which not only inhibits the production of ROS by the direct involvement of specific molecules [99], but also can modulate the Keap1-Nrf2/ARE pathway [100]. This is a powerful oxidation-reduction defense system, where polyphenols act to degrade specifically the Keap1 protein and regulate the Nrf2-related pathway [59,101].

Remila and co-workers [96] proved the high antioxidant capacity of PlL leaves extract using the oxygen radical absorbance capacity in macrophages, melanoma and mammary mouse cell lines. Furthermore, due to the activation of apoptosis mechanisms, the extracts significantly inhibited the growth of melanoma cells. The data was related to the richness in phenolics, flavonoids and tannins in the extracts. Similar conclusions were obtained by Atmani [94], who examined hexane and chloroform aqueous extracts of PlL leaves containing highly concentrated flavonoids. In relation to the peak of hydroxyl groups, the aqueous formulations strongly inhibited lipid peroxidation. The mechanism was explained by the scavenging of peroxyl radicals by the extracts. Radicals play a role in the development of cardiovascular disease and cancer. In this regard, the scavenger activity of gallic acid and galloylquinic derivatives, isolated from PlL leaves, is particularly attractive. Notably, a progressive increase in the anti-radical activity associated with the number of galloyl groups in quinic acid was found [102]. Furthermore, all the tested metabolites strongly reduced the oxidation of low-density lipoproteins, thus strengthening the protection of PlL against the lipid peroxidation.

In addition to having a preventive capacity, a potential ability to halt or reverse oxidative stress-related diseases has been attributed to PlL. In fact, the ability to fight aggressive tumors, i.e., neuro-blastoma [19], and other important tumor cell lines has been proved by in vitro studies [103,104].

Animal testing further explored the anti-ROS efficacy of PlL derivates. Ben Khedir and co-workers [70] determined the scavenger and anti-inflammatory activity of the fruits oil using the carrageenan-induced paw edema in a rat model. In that study, PlL oil demonstrated significantly better anti-inflammatory activity with edema inhibition, in comparison to those produced by the control NADPH. Moreover, PlL oil was able to increase the expression of superoxide dismutase, catalase and glutathione peroxidase, which are released as a response to the oxidative stress in the inflamed tissue. The effects were interpreted as a consequence of the content of humulene, caryophyllene and polyunsaturated fatty acid in the oil. Humulene and caryophyllene have been shown to inhibit the nuclear factor kappa B (NF-kB) pathway, responsible for the transcription of several proinflammatory cytokines, i.e., TNF-α, IL-1β, IL-6 and iNOS and COX-2 enzymes [105]. The polyunsaturated fatty acid in the oil might have partially replaced the arachidonic acid in the inflamed cell membranes [57], consequently lowering COX-2 production, the local inflammation and ROS generation.

Other in vivo studies demonstrated that an administration of PlL oil before the induction of the Bilateral Common Carotid Artery Occlusion followed by Reperfusion (BCCAO/R) was able to prevent the oxidative stress challenge in the nervous tissue due to the ischemic insult [12,106]. In the cerebral tissue, PlL oil restored the membrane phospholipid DHA and decreased the activity of the COX-2 enzyme. Additionally, PlL oil increased the concentration of the anti-inflammatory endocannabinoid congeners palmitoylethanolamide (PEA) and oleoylethanolamide (OEA) [12]. The outcomes were related to the high presence of the phytocannabinoid (E)-β-caryophyllene, which worked synergistically with the other compounds in the oil, expanding the levels of cannabinoid receptor type 2 (CB2) and PPAR-alpha receptors. Further studies attest to the role of β-caryophyllene as CB2 agonists, demonstrating its capacity to antagonize the release of cytokines from LPS-stimulated monocytes (TNF-α and IL-1 β) [7].

## 7. Potential Cytotoxicity

Starting from Paracelsus’ statement “the right dose differentiates a poison to a remedy”, investigations have been conducted with the intent to validate pharmaceutically PlL derivatives. For this reason, many studies have been conducted, in particular using in vitro and cell lines models testing different doses of PlL oil and extracts (Table 4). As a measure of risk versus benefit, many of them also applied enzymes testing, looking for the capacity of the EO and extracts to interact with the proteins and their involvement in inflammation and oxidative stress. In fact, it is well-known that the in vitro and cell lines systems are largely recommended to elucidate the safety of herbal products and propose the natural agents as nutraceuticals or xenobiotics [110]. Although such models lack the complexity of animals, and the compounds in testing should not exert in vivo the same effects as reported in isolated cell tissues [111], they have a significant role in predicting of risks and toxicology.

As a result of this work, it can be stated that not any published laboratory study reported cytotoxic effects which could be connected to PlL oil or extracts. Conversely, the experiments demonstrated direct or indirect biocompatibility of PlL derivatives to humans and non-human animals.

Notably, a wide range of biocompatibility was documented in oral cells. It was remarked specifically in oral human fibroblast cell lines using the EO from leaves and the MTT reduction assay [49]. That fact was further validated in the periodontal ligament fibroblasts, the gingival fibroblasts, the gingival keratinocytes and dysplastic oral keratinocytes applying the WST-1 metabolic activity assay [49].

The absence of side effects to oral cells is strengthened concerning the polyphenol extracts. In detail, the fruits extract showed high biocompatibility and a selecting index (SI) of cytotoxicity equal to > 256 toward the human gingival cells, at the same time demonstrating a strong response against periodontal bacteria [112].

## 8. Antimicrobial Activity

Several studies reported the antimicrobial activity of PlL oils and extracts, trying to clarify scientifically their popular use in infectious diseases.

Commonly studied pathogens comprise bacteria known for antibiotic resistance (*Staphylococcus aureus* incl. methicillin-resistant strains (MRSA), *Escherichia coli* and *Pseudomonas aeruginosa*), and other bacteria associated with the oral diseases, as well as yeasts, with particular regard to *Candida albicans* (Table 5a,b). 

Different antimicrobial activity was reported in the studies testing PlL EO and extracts. A high capacity against both bacteria and yeasts was demonstrated in regard to the leaves EO from plants growing in different regions [16,57,67]. Whereas the fruits EO from Tunisia [68] and Sardinia [58] were found to have a limited effect against bacteria and yeasts, the ethanol and water extracts of leaves from plants growing in Sicily inhibited the growth of *S. aureus, E. coli* and yeasts [107]. Additionally, the leaves ethanol or methanol extracts showed activity against several Gram-positive and Gram-negative bacteria [38,87,113]. However, leaves and stems methanolic or water extracts prepared in Algeria were inactive against *S. aureus* and *E. coli* [44], and leaves alcoholic extracts from Tuscany did not act against yeasts [109].

The research started from the fact that terpenoids in EO exert a wide-spectrum of antibacterial, antifungal and even anti-viral activity, and have been demonstrated to inhibit the growth of drug-resistant microbial strains, which are difficult to be treated even by conventional antibiotics [113]. Although α-pinene together with the monoterpenes terpinene and myrcene are among the most represented fractions in the Mediterranean oils showing antimicrobial capacity (Table 5), it is of general interest if higher activity could be related to specific PlL chemotypes [114]. In this context, as reported regarding the anti-inflammatory capacity, synergy between the chemical fractions of terpenes is proposed to explain PlL EO antimicrobial activities. Synergy was recently claimed to explain the inhibitory activity of the EO against *Porphyromonas gingivalis*, *Tannerella forsythia* and *Fusobacterium nucleatum* [49]. The result was related to the pharmacological interplay of the terpenes characterizing the oil chemotype from North Sardinia, which was α-pinene-terpinen-4-ol, further augmented by the non-cannabinoid terpenoids limonene and β-myrcene, the sesquiterpenes (Z)-caryophyllene and (E)- β- caryophyllene. Similarly, the antimicrobial efficacy of extracts can be attributed to the richness of polyphenols [25,38,46,69,70,76,111], in particular in regard to the concentrations of tannins, flavonoids, and lignin-carbohydrate complexes in the polyphenols mixture [11]. This is also the case with the ethanol extract of fruits, which showed the highest inhibition potential against *P. gingivalis* in comparison to 20 other extracts of pharmaceutical plants [112]. Furthermore, the potency of the fruits extract were higher than that of the leaves and woody parts, with MIC values against *P. gingivalis* of 8 μg/mL.

Recently, Mandrone and co-workers [108] related the antimicrobial activity of aqueous MeOH extracts of fruits and leaves to the concentration of phenolic components. That research further attested to the fact that phenols are active against multi-resistant bacteria, among them MRSA and carbapenemase-producing *Klebsiella pneumoniae*.

Concerning *Candida spp*., the activity of PlL derivates is reportedly controversial (Table 5). Low or no susceptibility of the yeasts to the leaves extract, or to the fruits oil, was found. Despite this, PlL leaves EO from Sardinia had low MIC values against *C. glabrata* and *C. albicans* [49]. The results were explained as a consequence of the recurrence of pharmacological concentrations of six terpenes, which were above 0.05% in the fingerprint. Additionally, the EO documented the ability to inhibit COX-2 and LOX, which are very important proteins for the development of *Candida* virulence [115].

Furthermore, a high inhibition of *C. albicans* was reported when using PlL extracts. The activity of water or ethanol extracts has been attributed to the flavonoid contents. Notably, the phenolic compound tannic acid was reported as more active against the yeast than the antifungals nystatin and amphotericin [68,75].

## 9. Summary and Conclusions

In this review, we summarized the existing knowledge about PlL phytochemistry and some of its biological activities, mainly focusing on its anti-inflammatory and antimicrobial capacity.

The chemistry shows that PlL EO is composed of up to 64 molecules, while 46 constituents have been identified in the extracts. Further minor fractions were determined in the reports analyzing both the EO and the extracts even if they could not be quantified. The more recurrent chemical components in the EO from plants growing in the Mediterranean area are represented by α-pinene, terpinenes, caryophyllene, limonene and myrcene. Important properties in antagonizing immune-mediated and autoimmunity, neuro-inflammatory, neurological and neurodegenerative diseases, in addition to infections and cancer have been addressed to these molecules. However, the biological character of PlL cannot be entirely focused on one of the main concentrated molecules. The abilities should be attributed to the whole mixture of the terpenes working in synergy, or in addition independently by their concentration in the agent. It is remarkable to note that concentrations of non-cannabinoid terpenoids equal to or above 0.05% increase the pharmacological potency of PlL oil.

Regarding the extracts, the high polyphenol content is attractive for prevention, and for therapy of chronic illness and infections. The richness in polyphenols suggests the use of the extracts as nutraceuticals in human health. Among the compounds, flavonol glycosides, i.e., myricetin and quercetin glycosides, could indicate a possible role of PlL in preventing diabetic complications and managing intestinal inflammatory response. Meanwhile, the concentration of bioactive flavanones should put forward the extracts to manage cholesterol absorption, glucose and lipid metabolism. Furthermore, the quinic acid, lignans and anthocyanins content is attractive in view of the high antioxidant capacity, in particular when water-extracted from PlL.

Nevertheless, although useful information has been provided in in vitro and animal model studies, clinical trials are necessary to fully understand the capacity and limitations of PlL EO and extracts in humans.

Regarding the anti-inflammatory capacity, PlL EO and extracts are able to inhibit the proinflammatory cytokines IL-1β, IL-6 and TNF-α. Additionally, the capacity to anatgonize the arachidonic acid cascade was highlighted in previous studies. In particular, the antagonism towards the COX-2 enzyme emerged in the research, and then the ability to antagonize the first phase of the active inflammation by inhibiting prostaglandins. Furthermore, the potential LOX inhibitory capability could propose the EO as a COX-2-LOX dual inhibitory natural compound, which might be promising against inflammation and tissue damage.

Among the non-cannabinoid terpenoid fractions, (E)-β-caryophyllene has been suggested to possess an important role in the anti-inflammatory activity of the oil. Interestingly, the molecule has a strong affinity to CB2, where it inhibits the release of cytokines from LPS-stimulated monocytes, such as TNF-α and IL-1 β expression. Furthermore, studies in animals have proved the relevance of (E)-β-caryophyllene in preventing ischemic/reperfusion oxidative injury when the oil was administered as a dietary assumption. In this context, studies strengthened the adjuvant capacity of α-pinene merged with caryophyllene, in PlL oils by the high anti-inflammatory properties. Conversely, when merged with the non-cannabinoid myrcene, caryophyllene contributed to inhibit nitric oxide production and IL-1β-induced iNOS mRNA, NF-kB and other catabolic and inflammatory mediators of importance in rheumatoid arthritis. Regarding the limonene fraction of the EO, it should be relevant in the oxidative stress-related diseases by inhibiting pro-inflammatory mediators, leukocyte migration and the vascular permeability.

The high antioxidant property of the extracts is remarkable in the literature. This fact has been explained by the richness of polyphenols, including tannins and flavonoids, and sterols showing high protection against free radical damage. Animal testing further supports the in vitro anti-ROS potency of PlL extracts, with a higher effect in comparison to NADPHs. Notably, the aqueous formulations show great capacity in inhibiting lipid peroxidation. The mechanism is interpreted as scavenging of peroxyl radicals, which suggests PlL to be preventive in cardiovascular disease and cancer. Furthermore, the antioxidant properties of polyphenols in PlL extracts should be beneficial in therapy and prevention of gastrointestinal diseases, where oxidative stress has been shown to damage the barrier protection, leading to intestinal pathologies. In this matter, the proved capacity of polyphenols to manage the microbial and metabolomic patterns in the body might represent an additional advantage. These facts should validate scientifically the popular use of PlL extracts to resolve infections and persistent inflammation at the intestinal tract, proposing the materials as natural agents in prevention and therapy of gastro-intestinal diseases.

In regard to the antimicrobial activity, the capacity of the oil and extracts against periodontal bacteria has been largely documented. These evidences can prove scientifically the popular use of PlL in the relief of gingival bleeding and tooth ache. The ability against periodontal bacteria, further ameliorated by the anti-inflammatory potency, is attractive with regards to a possible use of the EO to antagonize gingivitis, as a primary strategy to prevent periodontitis and as a secondary preventive strategy to prevent recurrent periodontitis after periodontal surgery. The possibility to formulate PlL derivates as potential oral health care products or therapeutics in periodontal disease is further strengthened by the largely documented biocompatibility and antioxidant capacity.

Other important considerations concern the activity of PlL against *Candida* infections. PlL inhibits the growth of *C. albicans* and *C. glabrata* with low MICs. In addition, the prevention of arachidonic acid oxidation by COX-2 and LOX antagonism by PlL oil should be of interest for inhibiting the development of *Candida* biofilm and disseminations. Subsequently, PlL could act directly against the yeast and indirectly against its virulence, with no oral cytotoxicity.

Taken together, all the above argumentations propose PlL as a nutraceutical, and also as a therapeutic agent against a wide range of diseases based on inflammation and infections. Further research may include the activity of PlL not only on planktonic microorganisms but also on biofilms. It should verify the best and most efficient preparation method of PlL plant material related to its activity.

## Figures and Tables

**Figure 1 antibiotics-10-00425-f001:**
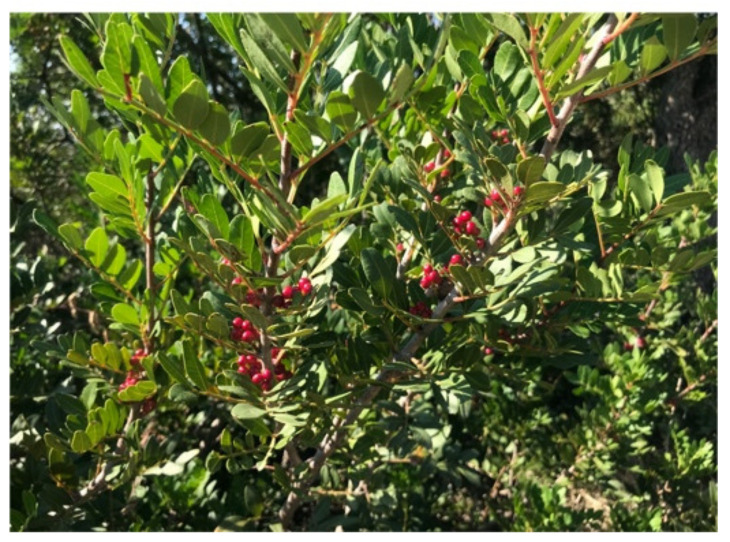
*Pistacia lentiscus* L., also known as mastic tree or lentisk.

**Table 1 antibiotics-10-00425-t001:** Taxonomy description (groups and species) of the *Pistacia* genus (adapted from Zohary [16]).

Group	Species
Lenticella	*Pistacia mexicana* HBK
	*Pistacia texana* Swingle
Eu lentiscus	*Pistacia lentiscus* L. (mastic tree)
	*Pistacia saporte* Burbar
	*Pistacia weinmannifolia* Poisson
Butmela	*Pistacia atlantica* Desf.
Eu terebintus	*Pistacia chinensis* Bge.
	*Pistacia khinjuk* Stocks
	*Pistacia palaestina* Bois.
	*Pistacia terebinthus* L.
	*Pistacia vera* L.

**Table 2 antibiotics-10-00425-t002:** Ethnopharmacological uses of *Pistacia lentiscus*
*L*.

Geographical Area	Ailment/Uses	Ref.
Sardinia, Italy	Oral cavity inflammation and infection, tooth ache, osteoarthritis, bronchitis, cough sedative, antipyretic, allergies, asthma, ulcerations, gastrointestinal disorders, wound healing and haemostatic	[19,25,26,27,28,29,30,31]
Southern regions of Italy (Calabria and Campania)	Inflammation of the mouth, tooth ache, mycosis, herpes and refreshing feet	[27,32]
Central regions of Italy (Abruzzi, Marche and Tuscany), Spain	Hypertension and cardiac diseases	[27,33]
Spain	Analgesic, teeth strengthening, hypertension and cardiac diseases	[33,34]
Tunisia	Antipyretic, astringent, eczema, paralysis, antimicrobial, throat infections, asthma, hypertension, cardiac diseases, paralysis, diuretic properties, renal stones, jaundice, antiatherogenic effect, antihepatotoxic and gastrointestinal diseases	[27,35]
Algeria	Stomach ache, dyspepsia, peptic ulcer, diarrhea and rheumatism	[36]
Morocco and North Africa	Hypertension, cardiac diseases and diabetes	[37]
Libya	Immuno-stimulant and antimicrobial	[23]
Jordan	Ameliorate jaundice	[38,39]
Israel	Heartburn and soothes stomach	[40]
Iran	Gum tissue strengthened, breath deodorizer, brain and liver tonic and gastrointestinal ailments	[41,42]
Turkey	Throat infections, asthma, eczema, stomach ache, renal stones, paralysis, diarrhea, jaundice, anti-inflammatory, antimicrobial, antipyretic, stimulant and astringent	[43]

**Table 3 antibiotics-10-00425-t003:** Chemical profiles of *Pistacia lentiscus* L.

Plant Material	Origin	Main Components of Essential Oils or Plant Extracts	Test Assays	Ref.
**Essential oils from**				
Leaves	Spain	β-myrcene (19%), α-terpineol + terpinen-4-ol (15%), α-pinene (11%)	GC-MS	[53]
Unripe fruit		β-myrcene (54%), α-pinene (22%),	GC-MS	
Ripe fruit		β-myrcene (19%), α-pinene (11%), δ-3-carene	GC-MS	
Leaves	Egypt	δ-3-carene (65%), sesquiterpene alcohols (4%)	GC-MS	[52]
Leaves	Greece	Myrcene (20.6%), germacrene D (13.3%), E-caryophyllene (8.3%), α-cadinol (7.3%), t5-cadinene (7.0%)	GC-MS	[62]
Leaves	Turkey	Terpinen-4-ol (29.9%), α-terpineol, (11.6%), limonene (10.6%), (*Z*)-3-Hex-1-enyl benzoate (6.7%), α-pinene (4.2%), β-caryophyllene (3.2%)	GC-MS	[63]
Leaves	Morocco	Myrcene (39.2%), limonene (10.3%), β-gurjunene (7.8), germacrene (4.3%), α-pinene (2.9%), muurolene (2.9%)	GC-FID; GC-MS	[54]
Leaves	Tunisia	α-pinene (16.8%), 4-terpinenol (11.9%), β-phellandrene (8.9%), sabinene (5.7%9, γ-terpinene (5.5%) and β-pinene (4.3%)	GC–MS	[55]
Aerial parts	Algeria (Algiers)	Longifolene (12.8%), γ-cadinene (6.2%), trans-β-terpineol (5%), α-acorneol (4.6%), γ-muurolene (4.2%), β-pinene (3.7%)	GC, GC-MS	[64]
	Algeria (Tizi-Ouzou)	Longifolene (16.4%), trans-β-terpineol (15.6%), terpinen-4-ol (7%), γ-muurolene (5.7%), β-pinene (3.3%), α-pinene (2.8%)	GC, GC-MS	
	Algeria (Oran)	α-pinene (19%), trans-β-terpineol (13.1%), sabinene (12.6%), β-pinene, (6.5%), (E)-β-ocimene (5.5%), longifolene (5.2%)	GC, GC-MS	
Leaves	Sardinia (Italy)	α-pinene (14.8–22.6%), terpinen-4-ol (14.2–28.3%), β-myrcene (1.0–18.3%), *p*-cymene (14.8–16.2%), sabinene (2.5–8.1%), limonene (0.9–3.8%)	GC-MS	[19]
Leaves	Greece	α-pinene (9.4–24.9%), terpinen-4-ol (6.8–10.6%), p-cymene (0.5–7.5%), limonene (9.0–17.8%), γ-terpinene (3.1–3.6%)	GC-MS	[51]
Leaves	Sardinia (Italy)	α-pinene, α-thujene, camphene, sabinene, β-pinene, myrcene, α-phellandrene, α-terpinene, para-cymene, β-phellandrene, trans-ocemene, γ-terpene, terpinolene, 2-nonanone, linalool, isopentyl isovalerate, terpin-4-ol, α-terpiniol and others.	GC-MS	[29]
Leaves	Algeria	β-caryophyllene (54–198 μg g^−1^ dw), δ-cadinene (15–186 μg g^−1^ dw), cubebol (15–117 μg g^−1^ dw), β-bisabolene (22.1- 105 μg g^−1^ dw), α-pinene (1.9–105 μg g^−1^ dw), γ-muurolene (29.7–67.3 μg g^−1^ dw)	GC-MS	[65]
Leaves	Sardinia (Italy)	Germacrene D (19.9%), β-caryophyllene (6.6%), α-pinene (6.3%), myrcene (3.9%), β-phellandrene (3.7%), α-humulene (2.4%)	GC-MS	[12]
Leaves	Eastern Morocco	Taforalt and Saidia areas: limonene, *α*-pinene, *α*-terpineol and *β*-caryophyllene;Laayoune and Jerada areas: myrcene and *β*-caryophyllene.	GC-MS	[66]
Fresh leaves	Greece	δ-germacrene (24.78%), myrcene (19.5%), α-cadinol (9.53%), γ-cadinene (5.59%), trans-caryophyllene (5.03%), limonene (4.84%)	GC-MS	[67]
Dried leaves		δ-cadinene (17.04%), α-amorphene (10.32%), δ-germacrene (9.01%), trans-caryophyllene (6.32%), α-cubebene (5.55%), naphthalene (4.13%)	GC-MS	
Ripe fruits	Tunisia	Phenolic composition of seed oil (concentrations not shown)	GC-MS	[68]
Leaves	Tunisia	Germacrene D (11.9%), pinene (9.9%), limonene (8.5%), δ-cadinene (8.5%), β-caryophyllene (8.2%), terpinen 4-ol (5.1%)	GC-FID, GC-MS	[69]
Fruits	Tunisia	α-pinene (13.35%), α-phellandrene (10.12%), β-phellandrene (10.45%), sabinene (7.01%), germacrene-D (6.86%), β-caryophyllene (4.58%)	GC-MS	[70]
Leaves	Tuscany (Italy)	α-pinene (24.6–9.2%), 1–4 terpineol (14.9–7.1%), β-phellandrene (11.4–4.7%), β-pinene (8.6–1.2%), β-mircene (9.2–0.7%), α-terpineol (8.4–4.9%)	GC-MS	[71]
Leaves and twigs	Sardinia (Italy)	Terpinen-4-ol (25.2%), α-phellandrene (11.9%), β-phellandrene (10.2%), γ-terpinene (10.1%), α-pinene (7.6%)	GC-FID, GC-MS	[72]
Fruits	Tunisia	4-{3-[(2hydroxybenzoyl) amino] anilino}4-oxobut-2-enoic acid (28.96%), β-myrcene (11.47%), 3-pentadecylphenol (8.51%), p-tolyl ester (8.36%), amino formic acid (7.51%)	GC–MS	[73]
Male flowers	Tunisia	β-caryophyllene (12.8%), germacrene-D (9.6%), elemol (8.9%), α-terpineol (7.8%), γ-cadinene (7.1%), bornyl acetate (6.2%)	GC-MS	[74]
Female flowers		α-limonene (28.7%), germacrene-D (23.7%), elemol (6.7%), β-caryophyllene (6.6%), α-pinene (6.0%), bornyl acetate (3.7%)	GC-MS	
Leaves of male plants at flowering		α-limonene (18.8%), germacrene-D (13.1%), β-caryophyllene (8.8%), δ-cadinene (8.7%), γ-cadinene (6.2%), α-pinene (4.8%)	GC-MS	
Leaves of female plants at flowering		Germacrene-D (20.7%), δ-cadinene (15.6%), β-caryophyllene (12.1%), γ-cadinene (6.6%), δ-cadinol (6.1%), α-limonene (5%)	GC-MS	
Ripe fruits		β-myrcene (75.6%), α-pinene (12.6%), α-limonene (3.2%), α-terpineol (1.4%), camphene (0.8%)	GC-MS	
Leaves	Morocco	Myrcene (33.5%), α-pinene (19.2%), limonene (6.6%), α-phellandrene (4.6%), γ-terpineol (3.7%), α-terpineol (3.6%)	GC-MS	B [75]
Leaves	Sardinia (Italy)	α-pinene (16.9%), terpinen-4-ol (16.5%), sabinene (7.8%), α-phellandrene (7.4%), γ-terpinene (6.3%), β-pinene (4.3%)	GC-MS	[49]
**Plant extracts/solvent used**				
Leaves/ethyl acetate and methanol	Italy	3,5-O-digalloyl quinic acid (26.8 ± 0.15 mg/g DW), 3,4,5-O-trigalloyl quinic acid (10.3 ± 2.45 mg/g DW), 5-O- galloyl quinic acid (9.6 ± 2.45 mg/g DW), myricetin 3-O-rhamnoside (6.8 ± 1.04 mg/g DW), myricetin 3-O-rutinoside (4.5 ± 0.18 mg/g DW), myricetin glucuronide (3.9 ± 0.65 mg/g DW)	HPLC-DAD, HPLC-MS, NMR	[60]
Berries/methanol	Apulia (Italy)	Cyanidin 3-O-glucoside (71%), delphinidin 3-O-glucoside, cyanidin 3- O arabinoside (28–31%)	HPLC-DAD-MS	[76]
Fruits during maturation/petroleum ether	Tunisia	Oils, fatty acids and sterols	GC-MS	[35]
Leaves/methanol	Algeria	46 compounds (most abundant flavonoids, phenolic acids and their derivatives)	HPLC-ESI-QTOF	[61]
Leaves/methanol	Italy	46 secondary metabolites	LC-ESI-MS/MS	[77]
Fruits/methanol-water	Tunisia	Total phenolic acids 436.4–2762.7 mg/kg; total flavones 75.3–1222.9 mg/kg; total flavonols 24.2–377.4 mg/kg; total secoiridoids 12.6–366.8 mg/kg; total phenols 538.0–4260.6 mg/kg	HPLC-DAD/MSD	[78]
Leaves/ethanol	Italy	Tannin derivatives (70.5%), myricetin derivatives (22%), quercetin derivatives (7.2%)	HPLC-DAD	[79]
Leaves/methanol	Egypt	α-pinene (38.1%), 3,5-O-digalloyl quinic acid (13.5%), D-limonene (11.9%), α-phellandrene (10.1%), β-pinene (9.5%), γ muurolene (8.0%), luteolin-3-O-rutinoside (7.8%), quercetin 3-O-di-hexose O-pentose (7.6%), 3,4,5-O-trigalloyl quinic acid (6.1%), quercetin 3-O-glucuronide (4.6%), epicatechin 3-gallate (4.5%), camphene (3.8%)	UHPLC-ESI-MS, GC-MS	[80]

**Table 4 antibiotics-10-00425-t004:** Antioxidant and anti-inflammatory activity of *Pistacia lentiscus.*

Exp. Setting	Origin Model	Plant Material	Model	Exp. Protocol	Results	Ref.
Antioxidant activity	Sardinia, Italy	Leaves oil	Cells free	DPPH as Trolox equivalent antioxidant capacity (TEAC)	Great seasonal variability inhibition	[19]
	Algeria	Leaves extract	Cells free	FRAP	↑ High	[86]
				H_2_O_2_ scavenging activity	↓ Low	
	Algeria	Leaves extract	Cells free	Ferric reducing antioxidant power (FRAP)	↑ High and dose dependent	[94]
				DPPH	↑↑ Very high	
				H_2_O_2_ scavenging activity	↑↑ Very high	
				Linoleic acid peroxidation inhibition	↑↑↑ Outstanding	
	Zakynthos (Greece)	Leaves extract	Cells free	DPPH	↑↑ Very high	[51]
				Ferric reducing antioxidant power (FRAP)	↑ High	
	Sardinia, Italy	Leaves extract	Cells free	DPPH as Trolox equivalents	↑↑ High	[95]
				ABTS as Trolox equivalents	↑↑ High	
	Algeria	Leaves extract	Cells free	DPPH (%)	↑ High	[35]
				Ferric reducing antioxidant power (FRAP)	↑ High	
				β-carotene bleaching method (%)	↑↑ Very high	
	Morocco	Fruits oil, leaves oil	Cells free	DPPH	Fruits oil: ↑↑ high Leaves oil: ↑ high	[75]
				FRAP	Fruits oil: ↑↑ high Leaves oil: ↑ high	
				ABTS	Fruits oil: ↑↑ high Leaves oil: ↑ high	
	Campania (Italy)	Leaves extract	Cell lines	Lipid peroxidation	↑↑ Very high	[15]
				Intracellular ROS	↑↑Very high	
				Oxidized glutathione	↑↑ Very high	
	Sardinia, Italy	Leaves oil	Animals	DHA	↑↑ High protection	[12]
	Algeria	Fruits extract, leaves extract	Cells free and cell lines	Intracellular ROS in THP-1 monocytic cells	Fruits extract: dose-dependent protection	[96]
				ORAC as μmol Trolox Equivalents	Fruits extract: ↑ High; Leaves extract: ↑↑ Very high	
	Sardinia, Italy	Leaves oil	Cells free and cell lines, human fibroblasts	H_2_O_2_ scavenging activity	↓ Low	[49]
				ECC	↓ Low	
Anti-inflammatory activity	Sardinia, Italy	Leaves oil	Animals	COX-2	↑↑ High inhibition	[12]
	Sardinia, Italy	Leaves oil	Animals	TNF-α	↓↓ High decrease	[29]
				IL-6	↓↓↓ High decrease	
	Algeria	Fruits extract, leaves extract	Cells free and cell lines	IL-1β inhibition by ATP stimulated THP-1	Fruit extract: no reduction; Leaves extract: ↑↑ high	[96]
				IL-1β inhibition by H_2_O_2_ stimulated THP-1	Fruit extract: ↓ low; Leaves extract: dose-dependent	
	Sardinia, Italy	Leaves oil	Cells free and cell lines, Human fibroblasts	COX-1	Inhibition	[49]
				COX-2	↑ high inhibition	
				LOX	no inhibition	

DPPH = 2,2-diphenyl-1-picryl-hydrazyl-hydrate assay; FRAP = ferric reducing activity power assay; MTT = 3-(4,5-dimethylthiazol-2-yl)-2,5-diphenyltetrazolium bromide; ABTS = 2,20-azinobis (3-ethyl- benzothiazoline-6-sulphonic acid) diammonium salt; WST-8 = (2-(2-methoxy-4-nitrophenyl)-3-(4-nitrophenyl)-5-(2,4-disulfophenyl)-2H-tetrazolium, monosodium salt) assay; IC50 = percentage of cytotoxicity and concentration which inhibit half-cell population compared to the drug-free control; CC50 = concentration which inhibits cell metabolism by 50% compared to the drug-free control; DHA = docosahexaenoic acid; BCB = β-carotene bleaching test; ECC = electro-chemical characterization assay; DCF = 2’,7´dichlorofluorescein diacetate; SRB = sulforhodamine B; ORAC = oxygen radical absorbance capacity; IL-1β = interleukin-1β; IL-6 = interleukin-6; TNF-α = tumor necrosis factor-α; COX-1 = cyclooxygenase 1; COX-2 = cyclooxygenase 2; and LOX = lipoxygenase.

**Table 5 antibiotics-10-00425-t005:** (**a**) Antibacterial activity of *Pistacia lentiscus* L. determined by agar diffusion test (ADD) or minimal inhibitory concentration (MIC). (**b**) Antifungal activity of *Pistacia lentiscus* L. determined by agar diffusion test (ADD) or minimal inhibitory concentration (MIC).

(a)						
Origin	Plant Material	Bacteria	Origin of Strain	Antimicrobial Activity		Ref.
Sicily (Italy)	Aerial parts ethanol extract, aerial parts water extracts	*Staphylococcus aureus*	ATCC 29213	Yes		[107]
		*Escherichia coli*	ATCC 35218	Yes		
Tunisia	Leaves essential oil	*S. aureus*	ATCC 25923	Yes		[55]
		*Enterococcus faecalis*	ATCC 29212	Yes		
		*Salmonella enteritidis*	ATCC 13076	Yes		
		*Salmonella typhimurium*	NRRLB 4420	Yes		
		*E. coli*	ATCC 25922	Yes		
		*Pseudomonas aeruginosa*	ATCC 27853	Yes		
Algeria	Leaves ethanol extract	*S.* *aureus*	ATCC 601	Yes		[86]
		*Listeria monocytogenes*	ATCC 19111	Yes		
		*Klebsiella pneumoniae*	5215773	Yes		
		*P. aeruginosa*	22212004	Yes		
		*S. typhi*	4404540	Yes		
		*Proteus mirabilis*	0536040	Yes		
		*E. coli*	5044172	Yes		
		*Enterobacter cloacae*	1305573	Yes		
			444	Yes		
Eastern Morocco		Aerial parts from different areas of Morocco essential oils	*S.* *aureus*	Not given	Yes	[66]
			*Streptococcus* spp.	Not given	Yes	
			*E. coli*	Not given	Yes	
			*K. pneumoniae*	Not given	Yes	
			*Pseudomonas* spp.	Not given	Yes	
			*Salmonella* spp.	Not given	Yes	
Tunisia		Fruits essential oil, phenolic extract	*S.* *aureus*	Not given	Yes	[68]
			*Bacillus subtilis*	Not given	Yes	
			*L. monocytogens*	Not given	Yes	
			*E. coli*	Not given	Yes	
			*P. aeruginosa*	Not given	Yes	
			*Aeromonas hydrophila*	Not given	Yes	
			*Salmonella* *typhimurium*	Not given	Yes	
Algeria		Aerial part methanol extract	*S.* *aureus*	Not given	Yes	[36]
			*E. coli*	Not given	Yes	
			*P. aeruginosa*	Not given	Yes	
Algeria		Leaves and stems methanol extract, leaves and stems aqueous extracts	*S.* *aureus* *E. coli*	Not given Not given	No No	[44]
Sardinia (Italy)		Fruits essential oil	*Bacillus clausii*	Probiotic	No	[58]
			*Staphylococcus* *hominis*	Clinical	No	
			*S.* *aureus*	ATCC 6538	No	
			*Streptococcus pyogenes*	Clinical	No	
			*Streptococcus agalactiae*	Clinical	Yes	
			*Streptococcus salivarius*	Probiotic (*n* = 2)	No	
			*Streptococcus mitis*	Clinical	No	
			*Streptococcus mutans*	Collection	No	
			*Streptococcus intermedius*	Collection	Yes	
Sardinia (Italy)		Fruit methanol extract, leaves methanol extract,	*S. aureus*	ATCC 25293	Yes	[108]
			*Staphylococcus* *epidermidis*	ATCC 12,228	Yes	
			*E. coli*	ATCC 25,922	In part	
			*K. pneumoniae*	ATCC 9591	In part	
Sardinia (Italy)		Leaves essential oil	*Streptococcus gordonii*	ATCC 10,558	Yes	[49]
			*Actinomyces naeslundii*	ATCC 12104	Yes	
			*Fusobacterium nucleatum*	ATCC 25586	Yes	
			*Porphyromonas gingivalis*	ATCC 33277	Yes	
			*P. gingivalis*	Clinical (*n* = 2)	Yes	
			*Tannerella forsythia*	ATCC 43300	Yes	
			*T. forsythia*	Clinical (*n* = 2)	Yes	
(**b**)						
**Origin**		**Plant Material**	**Fungi**	**Origin of Strain**	**Antifungal Activity**	**Ref.**
Sicily (Italy)		Aerial parts ethanol extract, aerial parts water extracts Leaves ethyl acetate and methanol extract	*Candida albicans* *Candida parapsilosis* *Candida glabrata* *Cryptococcus neoformans*	Clinical (*n* = 18) Clinical (*n* = 9) Clinical (*n* = 11) Clinical (*n* = 5)	Yes Yes Yes Yes	[107]
Tuscany (Italy)		Leaves ethyl acetate and methanol extract	*C.* *albicans*	Clinical	No	[109]
			*C.* *glabrata*	Clinical	No	
			*C.* *parapsilosis*	Clinical	No	
			*C. tropicalis*	Clinical	No	
			*C. zeylanoides*	Clinical	No	
Algeria		Leaves ethanol extract	*C.* *albicans*	444	Yes	[86]
Tunisia		Fruits essential oil, phenolic extract	*Aspergillus flavus*	Not given	No	[68]
			*Aspergillus niger*	Not given	No	
			*C.* *albicans*	Not given	In part	
Sardinia (Italy)		Fruits essential oil	*C* *. albicans*	Clinical	No	[58]
			*C.* *glabrata*	Clinical	No	
			*C.* *krusei*	Clinical	No	
Sardinia (Italy)		Leaves essential oil	*C.* *albicans*	Laboratory	Yes	[49]
			*C.* *albicans*	Clinical (*n* = 2)	Yes	
			*C.* *glabrata*	Laboratory	Yes	
			*C.* *glabrata*	Clinical (*n* = 2)	Yes	

## Data Availability

Data is contained within the article.

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
