# Peer review of "Leaves and Fruits Preparations of Pistacia lentiscus L.: A Review on the Ethnopharmacological Uses and Implications in Inflammation and Infection"

_antibiotics, 2021, doi:10.3390/antibiotics10040425_

Round 1
Reviewer 1 Report
Thank you very much for the opportunity of reviewing this paper. The topic of this review is interesting. Hovewer, it has important limitations.
First. The article is a narrative review and although the requirements for this type of work are not as high as for a systematic review, they still exist to assess the quality of the review.
So many relevant items are missing from this review, which makes the review low-quality and hard to judge its field contribution.
The most important are:
- Description of literature search is missing. It’s not necessary to describe the literature search as much detail as in systematic review, but it’s necessary to specify search terms, types of literature included, quantity and names of the databases searched.
- Data are not appropriate presented. In tables-in vitro study are presented together with in vivo. Results presented in the tables are different from this described in the main text.
- The best researched and known natural product from Pistacia lentiscus is resin (better known as mastic or mastix), however, the abstract lacks information that this article deals with other natural products: essential oil. leaf, fruit and wood extracts. Likewise, the title can be misleading.
- The conclusions drawn from the literature analysis do not match the data shown. There is an overinterpretation. Cox and Lox (paragraph 5) were examined only in one study, similar information about the NF-kB pathway, but authors wrote in the conclusions and abstract: “The anti-inflammatory activity mainly occurs due to inhibition of NF-kB pathway or directly toward the proinlammatory cytokines, or arachidonic acid cascade against COX-2 and LOX "...???
- There is chaos in the description of the results and the attitude to the results in the tables. I am unable to determine the value of this work without appropriate corrections.
- References in the tables are provided in a different style than in the main body of the manuscript.
- proposed table 2 style:
|
Geographical area |
Ailment/uses |
part of the plant and type of natural product used
|
Author |
References |
|
Sardinia, Italy
|
Oral cavity …
|
Essential oil from leaves..? or water extracts from leaves and so on. |
Camarda 1990
|
[2].. |
- The organization of table 4 should relate to the remaining tables, i.e. the author, the last reference number. In the plant material column, I often lack information on the type of extract - whether aqueous, methanol or mixed ..?
- In table 4. Anti-inflammatory activity of Pistacio lentiscus. There are mainly data about antioxidant activity of P. lentiscus. Despite the fact that the antioxidant properties are often correlated with anti-inflammatory activity, it cannot be said that the antioxidant activity is indicative of the anti-inflammatory properties of the extract, as is shown in the article and in Table 4.
- Line 89- where is figure 1?
- Reading the manuscript I have the feeling that the authors favor essential oils and terpenoids in interpreting the data. The collected data should be analyzed objectively. They show that other compounds play an important role, e.g. tannins.
Author Response
We thank the reviewer for his/her critical review. We tried our best to completely rewrite the manuscript and followed most of the suggestions.
- “Description of literature search is missing. It’s not necessary to describe the literature search as much detail as in systematic review, but it’s necessary to specify search terms, types of literature included, quantity and names of the databases searched.”
We agree that it is in part misleading because of the too many added detailed information. But as the reviewer states correctly, this is a narrative review. There it is not common to present the exact search criteria. However, we have added the point. Thank you.
- Data are not appropriate presented. In tables-in vitro study are presented together with in vivo. Results presented in the tables are different from this described in the main text.
We checked all the tables and the text. Thank you.
Reviewer:
- “The best researched and known natural product from Pistacia lentiscus is resin (better known as mastic or mastix), however, the abstract lacks information that this article deals with other natural products: essential oil. leaf, fruit and wood extracts. Likewise, the title can be misleading.”
The reviewer is right that we excluded mastic from our search. We clarified that fact now in the title, abstract and main text. Thank you.
Reviewer:
- “The conclusions drawn from the literature analysis do not match the data shown. There is an overinterpretation. Cox and Lox (paragraph 5) were examined only in one study, similar information about the NF-kB pathway, but authors wrote in the conclusions and abstract: “The anti-inflammatory activity mainly occurs due to inhibition of NF-kB pathway or directly toward the proinlammatory cytokines, or arachidonic acid cascade against COX-2 and LOX "...???”
We carefully revised the whole manuscript which included the conclusions.
- There is chaos in the description of the results and the attitude to the results in the tables. I am unable to determine the value of this work without appropriate corrections.”
- References in the tables are provided in a different style than in the main body of the manuscript.
The style of the references was checked and adapted. Thank you.
Reviewer 2 Report
Dear Authors!
I marked in yellow my point of view regarding the manuscript. I appreciated that first column is nota necessary in the table 1.
Write the genus for each species mentioned in the table 5.

Author Response
Dear Review,
just a note to thanks you for making our manuscript more clear and interesting.
Regarding table 1, we have corrected it as you kindly have indicated.
Regarding table 5, we have completely edited.
Thanks again for your comments and suggestions.
Reviewer 3 Report
The manuscript submitted for review is a review of the anti-inflammatory and antimicrobial properties of Pistacia lentiscus.
The publication is an interesting study, however, it requires minor changes that will facilitate the analysis of the presented results
The authors did not present what database they used, how many works were compared, and what years they were for.
The keywords are inaccurate and need to be corrected. For example why only Candida and periodontal bacteria and not others.
In the work, the names of microorganisms should be written in italics.
It is necessary to redraft and change the readability of the table:
In table 2 with references only numbers without authors
Table 3, not the authors, only references and only a number
Use a different layout in Table 4: main products, plant material, plant origin, experimental setting, studied variable, results, references
Table 5 requires major changes. I suggest the following layout: test material, main components, origin, test methods, results (all results described together) and references.
Results and summary described correctly.
The manuscript may be ready for publication with minor corrections.
Author Response
Dear Reviewer thank you for that statement.- The authors did not present what database they used, how many works were compared, and what years they were for.
This is not a systematic review and following information about literature search can be limited. However we added the used data bases.
- The keywords are inaccurate and need to be corrected. For example why only Candida and periodontal bacteria and not others.
We revised the keywords and excluded special microorganisms.
- In the work, the names of microorganisms should be written in italics.
The reviewer is right.
- It is necessary to redraft and change the readability of the table:
In table 2 with references only numbers without authors
Table 3, not the authors, only references and only a number
Use a different layout in Table 4: main products, plant material, plant origin, experimental setting, studied variable, results, references
Table 5 requires major changes. I suggest the following layout: test material, main components, origin, test methods, results (all results described together) and references.”
We reorganized all the tables. Thank you.
- Results and summary described correctly.
The manuscript may be ready for publication with minor corrections.
Thank again.
Reviewer 4 Report
Dear Authors,
After the review process, I have several comments: you should clearly present the aim of the paper in the abstract and final part of the introduction sections; you should include more recent data, in the introduction; you should have a justification of how and why natural products and their biomolecules are efficiencies in the management of anti-inflammatory and antimicrobial capacity; Please refer to: https://doi.org/10.3390/biomedicines8020039; the authors should modify the references in tables – name with reference number; the authors should present a limitation of the study and possible toxicological effects after administration; in general, the aspect of the table are too hard to be read, they should be simple and clear; in my opinion, they contain too much research data, that are not possible to be correlated of the readers.
Best regards!
Author Response
We thank the reviewer for the statement and we read with interest the recommended manuscript although our focus was on preparations of a defined plant and not on extracting and using biofunctional compounds. We tried to present the data in a more focused way also by deleting some information (the interested reader can read the reference). And we strengthened the aspect on future research.
Thanks again.
Round 2
Reviewer 1 Report
The manuscript was changed correctly and I suggested accepting it.
One note: Abbreviation “ PlL EO” are used in the abstract without earlier explanation.
Reviewer 4 Report
Dear Authors,
I do not have other comments.
Best regards!